# Altering the Hydrogen Isotopic Composition of the Essential Nutrient Water as a Promising Tool for Therapy: Perspectives and Risks

**DOI:** 10.3390/ijms26094448

**Published:** 2025-05-07

**Authors:** Nataliya V. Yaglova, Sergey S. Obernikhin, Ekaterina P. Timokhina, Elina S. Tsomartova, Valentin V. Yaglov, Svetlana V. Nazimova, Marina Y. Ivanova, Elizaveta V. Chereshneva, Tatiana A. Lomanovskaya, Dibakhan A. Tsomartova

**Affiliations:** 1Laboratory of Endocrine System Development, A.P. Avtsyn Research Institute of Human Morphology of Federal State Budgetary Scientific Institution “Petrovsky National Research Centre of Surgery”, 119991 Moscow, Russia; ober@mail.ru (S.S.O.); rodich@mail.ru (E.P.T.); tselso@yandex.ru (E.S.T.); vyaglov@mail.ru (V.V.Y.); pimka60@list.ru (S.V.N.); 2Department of Human Anatomy and Histology, Federal State Autonomous Educational Institution of Higher Education I.M. Sechenov First Moscow State Medical University, 119435 Moscow, Russia; ivanova_m_y@mail.ru (M.Y.I.); yelizaveta.new@mail.ru (E.V.C.); tatiana_80_80@inbox.ru (T.A.L.); dtsomartova@mail.ru (D.A.T.)

**Keywords:** water, deuterium-depleted water, hydrogen, isotope effect, oncology, neurology, immunology, endocrinology, anemia

## Abstract

Water is a vital nutrient that is needed to maintain almost all biological processes in living organisms. The natural water contains two isotopes of hydrogen—protium and deuterium. Deuterium, the trace component of natural water, significantly changes its physical and chemical properties and biological action. In this review, the authors summarize data on the isotopic effects of deuterium and discuss the possible magnetic effects of isotopes and the molecular basis of the biological effects of deuterium-depleted water. The review also presents new data on the already known and potential use of deuterium-depleted water in medicine (oncology, neurology) and previously unknown new directions of its use (immunological and endocrine disorders, anemia). Based on the analysis of collected data, the authors presented mechanisms of action of deuterium-depleted water in the organism. The authors also identified the least-studied effects of deuterium-depleted water, particularly its influence on morphogenetic processes. The review discloses the perspectives on deuterium-depleted water as a tool for therapy and substantiates the risks of its uncontrolled intake.

## 1. Introduction

Water is a vital nutrient that is needed to support all biological processes in living organisms. It is the major constituent of cells, tissues, and organs, ensuring their structure and functionality. Water is a universal solvent that allows the transport of nutrients, oxygen, and biologically active compounds to cells and the removal of metabolic products. The high heat capacity of water is of great importance for thermoregulation, providing stability of body temperature and preventing overheating and hypothermia. Water is involved in virtually all metabolic processes, and maintaining an optimal water balance is essential for the normal functioning of all organs and systems. Two main sources of water supply to organisms are known: external and internal. The external source is ingested natural water, and the internal one is intrinsic water produced in metabolic reactions. Several isotopic forms of water are present in prokaryotic and eukaryotic cells [1,2]. The ratio of different isotopic forms of water is a stable parameter for living cells. Changes to this ratio can lead to various functional changes due to isotopic effects. In this review, we have considered the major isotopic forms of water, its transportation into cells, isotopic effects arising from heavy and light isotope substitution, and the main clinical uses of water with modified hydrogen isotope content. Also, the authors formulated the most probable mechanisms of deuterium-depleted water action based on the analysis of these data.

## 2. Isotopic Forms of Water

The most common isotopic form of water is ^1^H_2_^16^O. It is a major solvent for most biological molecules and an agent of all biochemical reactions.

Heavy water (D_2_O or ^2^H_2_^16^O) is distinguished by the presence of a neutron in the nucleus of the hydrogen atom. Such an atom is called a deuteron, and the molecule is called deuterium. The mass of deuterium is twice that of protium. D_2_O differs from H_2_O in some physical properties such as viscosity, density, specific gravity, melting and boiling points, and the deuterium bonds are much stronger and shorter than the hydrogen bonds in H_2_O [3,4]. This means that the properties of heavy water as a solvent are also different and therefore change the kinetics of chemical reactions. But it is almost never found in nature. This is due to the exchange of hydrogen atoms between D_2_O and H_2_O, which results in the formation of semi-heavy water.

Semi-heavy water (HDO or ^1^H^2^H^16^O) is the most common form of deuterated water in nature [2,5,6]. Like hydrogen, oxygen has stable isotopes too. Water containing the oxygen isotope ^18^O instead of the most common ^16^O (^1^H_2_^18^O) has been found in biological systems [7,8]. It is considered to have a negligible effect on biological processes and is used as a tracer in metabolic studies [9,10].

## 3. Transportation of Water into Cells

Water is a major component of all human cells and tissues, and the same applies to all other vertebrates, invertebrates, single-celled organisms, and plants. The phenomenon of cell membrane water permeability has been debated by physiologists and biophysicists for decades. Most scientists agreed that water passes through biological membranes by simple diffusion through the lipid bilayer. The plasma membrane of any cell type is a major barrier to the movement of water between cells, but identifying the molecular pathways by which water is absorbed and released from cells remained unknown until the classes of membrane transport proteins were discovered.

The discovery of a 28 kDa integral membrane protein in erythrocytes and renal tubules ended the debate about the possible existence of molecular water channels [11]. The protein, now known as aquaporin 1 (AQP1), was discovered in erythrocyte membranes by the G. Benga group in 1985 and first isolated and purified by the P. Agre group in 1988 [11,12]. AQP1 was found to exist as a tetramer with intracellular N- and C-termini (Figure 1), an organization similar to that of several ion channel proteins [13,14]. Unit water permeability was found to be extremely high, ~3 × 10^9^ water molecules per subunit per second (i.e., ~450,000 deuterium molecules), while transport of other solutes or ions (including protons) was negligible [15,16]. Notably, studies on oocytes and reconstructed membranes confirmed that simple membranes exhibit finite water permeability, but the water permeability of membranes containing AQP1 is up to 100-fold greater. Thus, both sides in the long-standing controversy about water transport through membranes by simple diffusion or through water channels were right. In this connection, even slight changes in the deuterium content in water can be assumed to significantly affect various metabolic processes in the cell [17]. Moreover, the isotopic composition of water has been shown to influence the expression of aquaporins and thereby regulate water entry into cells. Thus, in the study of hepatic stellate cell proliferation, it was found that D_2_O treatment decreased the expression levels of AQP11 [18].

## 4. Isotopic Effects

Isotope effects are differences in the properties of an element due to the different masses of its isotopic atoms. The unequal atomic weights of isotopes cause certain differences in such properties of isotopic compounds as density, viscosity, refractive index, diffusion coefficient, specific charge of ions, etc. The change in energy levels during isotopic substitution, in turn, causes changes in thermodynamic properties such as heat capacity, thermal conductivity, heats of vaporization and melting, and boiling and melting temperatures. The chemical properties of isotopic compounds, meanwhile, remain largely unchanged, since the mass of the atom does not affect its electronic configuration, which determines the chemical properties. However, the thermodynamic inequality of isotopic compounds leads to an uneven distribution of isotopes at isotopic exchange equilibrium (thermodynamic isotope effect) [19]. In addition, the thermodynamic inequality of the initial isotopic compounds in combination with a similar inequality of transition states (active complexes) in the chemical reactions of isotopic compounds causes a difference in the rates of these reactions (kinetic isotope effect). Thus, the replacement of protium by deuterium in chemical bonds significantly affects the rate of bond breaking and the relative rate of chemical reactions. Among the spectrum of kinetic isotope effects, primary and secondary effects can be distinguished. More significant isotope effects are observed when the deuterium-containing bond is localized directly at the reaction site. In such cases, they are referred to as primary isotope effects. Deuterium atoms located near but not in the reaction center can cause secondary isotope effects, which are usually much weaker than primary effects [20].

The key to interactions between chemical elements, in particular between radicals that are spin carriers, is spin conversion driven by magnetic interactions. The nuclear spin system influences the magnetic behavior of the electron spin system and, ultimately, the reactivity of the chemical interaction. The nuclear spin selectivity of chemical reactions results in a difference in the reaction rates of radicals containing magnetic and non-magnetic nuclei. This phenomenon is called the magnetic isotope effect, in contrast to the well-known classical isotope effect, which is based on the nuclear-mass selectivity of chemical reactions [21,22]. Some isotopes, such as 1H, 13C, 15N, 25Mg, 31P, and 67Zn, are characterized by the presence of nuclear magnetic moments, or nuclear spins, and are called magnetic. Magnetic isotope effects in water molecules are poorly understood and are of considerable interest in identifying potential biological effects.

## 5. Molecular Basis for the Biological Effects of Deuterium-Depleted Water

Body deuterium content varies in humans. Usually, the concentration of deuterium in the organism is 12–14 (mM/L), i.e., deuterium is one of the first among the chemical elements in the body in terms of quantity [23]. Deuterium concentrations in normal fresh water range from 135 to 158 ppm [16]. The isotopic composition of water in the human body may vary depending on the geographical location of the place of residence and the source of consumption (artesian, glacial, or mineral springs). D/H ratios in blood and tissues may fluctuate depending on the diet. For example, lipid-based diets can increase intracellular water production by about twice as much as carbohydrate-based diets, which occurs through cooperative mitochondrial and peroxisomal lipid-substrate oxidation to H_2_O [24,25,26].

Some scientific reports show a lower deuterium content in natural water in earlier geologic periods compared to the present [27]. This observation became the basis for the hypothesis that a decrease in deuterium concentration may have played a key role in accelerating the evolutionary development of living organisms. It is assumed that water containing less deuterium has a stimulative effect on various biological processes, like optimization of metabolism, acceleration of cell growth and maturation, and increase in resistance to stress factors. Deuterium-reduced water can also affect genetic activity, protein synthesis, and energy metabolism in cells [28,29].

There are several theories explaining the stimulating effect of deuterium-reduced water on various systems in the body. The first theory is associated with differences in the rate of chemical reactions and the above-mentioned kinetic isotopic effect. The replacement of protium by deuterium in chemical bonds significantly slows down the rate of bond breakage and the relative rate of chemical reactions. At the same time, the reverse substitution significantly accelerates biochemical reactions in the organism [20].

The second theory concerns the provision of energy required for metabolic reactions. Chemiosmosis is a biochemical mechanism by which the energy of the electron transfer chain is converted into adenosine triphosphate (ATP) energy. Chemiosmosis utilizes membrane potential and an ion gradient, usually a proton gradient, to generate ATP. The electron transfer chain in mitochondria carries protons into the space between the inner and outer mitochondrial membrane. The protons then pass through ATP synthase, generating ATP [30]. It has been shown that the proton forms a covalent bond with the previously hydrogen-bonded oxygen atom of a neighboring water molecule. This water molecule then gives up another covalently bound proton, which binds hydrogen to the oxygen of a neighboring water molecule. The process continues, creating a kind of water “conductor wire”. This phenomenon has been called the Grotthuss effect. It was shown that the Grotthuss mechanism operates when water passes through ATP synthase [31]. Even a small enrichment of the medium with deuterium leads to reversible metabolic arrest and slowed chemiosmosis. It was even suggested that one of the functions of mitochondria is the generation of deuterium-free water, i.e., depletion of naturally consumed deuterium by the organism, thus optimizing the efficiency of metabolic processes in mitochondria [25,31,32,33]. The study demonstrated the regulation of macroenergy compound production in cells with impaired proton transport in mitochondria by reducing the deuterium concentration in water corresponds with this theory and shows the possibility of correcting cell growth, division, and functional abnormalities [34].

In addition to the direct influence of deuterium on the rate of chemical reactions due to the difference in molecular mass with protium, i.e., chemical and biochemical mechanisms of influence on biological processes, there is a hypothesis about physiological effects of changing the balance of deuterium and protium in the organism. It states that a shift in the balance of the stable hydrogen isotopes exerts a stimulative effect on the nonspecific defense of the organism due to the appearance of a concentration gradient of deuterium and protium. It is known that reduction in deuterium content after consumption of deuterium-depleted water in organs and tissues occurs at different rates. This is due to the formation of an isotope gradient and intensification of D/H-exchange reactions, accompanied by a significant decrease in deuterium content first in blood plasma and then in liver, kidney, and heart tissues. The most active reactions of D/H exchange take place in compounds that contain atoms with an unshared electron pair and are capable of forming intermediate reaction complexes with the participation of hydrogen bonds, in which a synchronous transition of protons and deuterons from one molecule to another is realized. Therefore, such exchange is more often observed in compounds having hydroxyl, less often thiol groups, and primary and secondary amino groups. Unlike oxygen, sulfur, and nitrogen, hydrocarbon bonds do not exchange protons and deuterons in natural conditions. That is why complete isotopic D/H exchange in living cells, where most hydrogen atoms are associated with carbon atoms, is practically impossible [25,35]. Thus, the substitution of deuterium by protium occurs mainly in highly reactive groups, which allows the metabolic rate to change. Altered metabolic patterns suggest physiological processes are affected.

Thus, changes in deuterium concentration in the consumed water can provide a tool for creating a mechanism to finely regulate physiological functions [36]. At the same time, it cannot be overlooked that a strong decrease in body deuterium content is a stress factor, since evolutionarily the organism is accustomed to a natural hydrogen isotope abundance [37].

## 6. Possible Applications of Water with Modified Hydrogen Isotope Composition in Medicine

Among the medical fields in which deuterium-depleted water has been used, oncology accounts for the largest amount of data. However, the use of deuterium-depleted water has expanded in recent years to include a wide variety of pathologies.

### 6.1. Applications in Oncology

As one of the leading causes of lethality worldwide, malignant tumors are responsible for almost 10 million deaths per year [38]. Currently, there are various therapies for malignant neoplasms, such as surgery, radiation therapy, chemotherapy, targeted therapy, and cellular immunotherapy. Although chemotherapy and radiation therapy can significantly prolong the life of patients, numerous side effects, including immunodeficiency, alopecia, diarrhea, fatigue, anorexia, anemia, etc., can significantly impair their quality of life [39]. Deuterium-depleted water has been found to significantly enhance the effect of chemotherapeutic drugs and reduce side effects and therefore can serve as an adjuvant agent in the treatment of tumors [28]. Modified water with a 25–125 ppm deuterium concentration is commonly used in clinical practice to reduce body deuterium content. Studies show that deuterium-depleted water inhibits the proliferation and migration ability of different tumor cell types, including lung, nasopharyngeal, breast, colorectal cancer, Ehrlich tumors, and pancreatic tumors [40,41,42,43,44,45,46]. These studies showed that deuterium-depleted water affects the cell cycle by decreasing the number of cells in the S phase and significantly increasing in the G1 phase and reduces oxidative stress by induction of antioxidant enzymes. Deuterium reduction also accelerates apoptosis, autophagy, and senescence in tumor cells. Of particular interest, however, is the fact that instead of inhibiting growth, deuterium-depleted water promoted the growth of normal cells [47,48]. However, in a single report, it was shown that deuterium reduction has no significant effect on cancer cells [43]. However, combinations of deuterium-reduced water with standard antitumor regimens show striking increases in proliferation inhibition, cell cycle arrest, and abnormal production of reactive oxygen species [31,40,41,44,49,50,51]. The use of deuterium-depleted water in combination with standard therapy can reduce the doses of chemotherapeutic drugs to achieve the same desired therapeutic results, thereby alleviating the side effects of chemotherapeutic drugs, as well as the cost of treatment.

How much deuterium reduction is required in water to establish optimal adjuvant treatment protocols is still unknown. The selection of deuterium concentrations and the duration of water consumption, as well as the timing of water consumption initiation, may be critical to successful tumor therapy and even prevention of metastasis. In our previous experiments, we have shown that consumption of deuterium-depleted water concurrently with tumor development has no significant effect on tumor growth and metastasis. But preliminary consumption of such water had a pronounced effect on inhibition of tumor growth and development of distant metastases [52]. Thus, changing the hydrogen isotopic composition of water is a viable factor to regulate tumor growth. However, its use should be based not on empirical data and phenomenological effects but should be the result of scientific studies revealing the mechanisms of antitumor effects and the time of their development.

### 6.2. Restoration of Immune Function

Besides the above-mentioned mechanisms by which deuterium-depleted water suppresses tumor growth, the restoration of immune functionality seems to be an additional sanogenetic effect observed in vivo. The immune system provides recognition and elimination of tumor cells, but a decrease in immune function with age or after exposure to external negative factors may result in the failure of immune control. An important role in antitumor immunity is played by the thymus as a central organ of the immune system producing T cells responsible for tumor cell elimination [53].

Investigations of the long-term decrease in deuterium level in the organism have revealed that morphogenetic processes providing thymopoiesis are sensitive to deuterium content in blood and tissues. Short-term consumption of deuterium-depleted water resulted in a boost in the activation of T cell production and changed the balance of T helper and T cytotoxic lymphocytes in the thymus, while a prolonged reduction in body deuterium content returned the ratio of T-lymphocyte subpopulations to the control values and, therefore, did not cause significant changes in the balance of T helper and T cytotoxic lymphocytes. At the same time, long-term consumption of deuterium-depleted water provided a higher content of proliferating blasts in the subcapsular region of the thymus cortex. Taken together, the data obtained indicate a slowing down of age-related involution of the thymus [54]. These results allow us to consider deuterium-depleted water as a possible tool for restoring immune system function that declined due to age-related factors or anthropogenic influences.

### 6.3. Antianemic Effect of Deuterium-Depleted Water

Anemias are a group of diseases manifested by lower red blood cell number and hemoglobin content due to impaired erythropoiesis and decreased synthesis of hemoglobin. Anemias result from either nutritional disorders or chronic diseases, as well as treatment with cytostatic drugs, radiotherapy, etc. The main causes of anemic disorders are iron and vitamin deficiency and a low-protein diet. Anemia is a major public concern since it originates from a low standard of living. One-fourth of the global population is estimated to be anemic, with cases increasing rapidly for women, expectant mothers, young girls, and children younger than age 5 [55,56,57].

Our studies have shown that gradual replacement of deuterium by protium by long-term consumption of 10 ppm deuterium-depleted water stimulates erythropoiesis in healthy rats. This effect was achieved without any additional external influence on erythropoietic factors, indicating an important role of deuterium in the regulation of hematopoiesis [58].

The mechanisms by which deuterium affects erythropoiesis include activation of hemoglobin synthesis, the main protein of erythrocytes that provides oxygen transport. At the same time, deuterium content in the organism acts as a regulatory factor affecting the rate of proliferation and differentiation of erythroblasts—young erythrocyte progenitors. Hemoglobin production is known to be tightly dependent on iron content. Deuterium/protium water disbalance seems to exert some effect on iron metabolism. Presumably, a gradual replacement of deuterium by protium improves assimilation of iron from food and makes the incorporation of iron into hemoglobin more efficient. This indicates that deuterium can regulate not only erythropoietic processes, that is, proliferation and maturation of red blood cell progenitors, but also their metabolic support [58].

The substitution of deuterium by protium in water also influences erythrocyte membranes. Consumption of water with a deuterium concentration of 10 ppm for 60 days has been shown to stimulate the antioxidant defense system of erythrocytes by reducing lipid peroxidation, preventing their premature destruction, thus providing an additional antianemic effect [59].

These data indicate that the hydrogen isotopic composition of water plays an important role in regulating the rate of erythropoiesis and iron metabolism, which is essential for maintaining the normal functioning of the hematopoietic system and providing the organism with oxygen.

### 6.4. Neurotropic Effect of Deuterium-Depleted Water

Water has never been considered a neurotropic agent. Investigations of deuterium-depleted water effects in vivo have found its ability to alleviate depression and anxiety, as well as improve long-term memory. The positive results were confirmed in experiments on laboratory mice that consumed water with a deuterium concentration of 90 ppm. The mice, after two weeks of deuterium reduction, showed milder symptoms of depression after exposure to chronic stress. The authors attributed this effect to changes in serotoninergic neurotransmission [60]. The sensitivity of the central nervous system to changes in the isotopic composition of water was also confirmed in experiments on Wistar rats, which were given water with deuterium content depleted to 20–25 ppm. Compared to the control, the treated rats showed less fear and anxiety in unfamiliar environments [61]. In another study conducted by the same authors, Wistar rats consuming deuterium-depleted water showed improved long-term memory compared to animals consuming water with standard deuterium concentration [62]. Later, another group of researchers obtained data on the increase in neuronal resistance to glucose deprivation and hypoxia with reduced body deuterium content, largely explaining the mechanisms of the above-described functional changes in rats [63].

The neuroprotective effect of deuterium-depleted water on humans has also been shown in epidemiological investigations. T. Strekalova et al. investigated the relationship between the incidence of depression in the USA and the deuterium content of tap water and found a strong positive correlation between the two. It was estimated that the prevalence of depression increases 1.8% for every 10 ppm increase in deuterium in tap water [60]. However, it should be noted that the level of anxiety and depression in the population depends largely on social and economic factors.

The mechanisms underlying the anxiolytic and nootropic effects of deuterium-depleted water are still poorly understood in spite of the facts elucidating certain metabolic and functional changes like ion transportation, fluidity of cell membranes, or synaptic activity. The results obtained in the experiments do not quite fit into the existing concepts about the properties of deuterium. Although the mechanisms of neuroprotective action are not well understood, deuterium-reduced water may be considered for use in monotherapy or in combination with other drugs to improve cognitive performance, especially in the elderly.

### 6.5. Possible Applications of Deuterium-Depleted Water in Endocrinology

The endocrine system controls all somatic functions by regulating cell metabolism, proliferation, differentiation, and apoptosis, and its dysfunctions lead to multiple disorders. That is why the growing number of endocrine and metabolic disorders is a worldwide public health problem. Implementation of deuterium-depleted water in endocrine practice is an open question, as the available data on the effects of deuterium depletion on endocrine gland function are small and scattered. They are not yet systematic, and therefore, there is no complete understanding of what kind of effects can be expected from the use of deuterium-depleted water.

Direct effects of deuterium depletion on hormone secretion have not been identified yet, but some reports do demonstrate alterations in endocrine function induced by consumption of deuterium-depleted water. It is well known that the endocrine glands are differentiated by the type of regulation of their function. Some of them, like the pancreas and the parathyroid gland, depend on metabolic parameters and change functional activity to normalize concentrations of certain substrates. Consequently, deuterium-depleted water can change the functioning of these glands both by affecting the metabolism of substrates and hormonogenesis in the gland itself. Other endocrine glands (the thyroid, adrenals, ovaries, and testicles) are governed by the hypothalamus–pituitary complex and operate via feedback loops. It means that besides directly affecting the secretory process, deuterium-depleted water can interfere in the regulation of endocrine function by the key regulators. Among the endocrine glands, the pancreas and the thyroid draw more attention from researchers since diabetes and thyroid disorders are the most common endocrine diseases affecting millions of people [64].

The thyroid is a typical gland with hypothalamus–pituitary regulation. Unlike other glands, the thyroid gland has a complicated secretory cycle. Thyroid cells form follicles with cavities where iodine atoms attach to thyroglobulin molecules, and the synthesized prohormone is stored. As needed, it is resorbed by the cells, where proteolytic cleavage of the prohormone to the active forms of the hormones thyroxine and triiodothyronine occurs. Our study of the thyroid and pituitary gland in young, sexually mature male Wistar rats showed that a reduction in body deuterium content by 10 ppm deuterium-depleted water provoked a significant increase in thyroid hormone secretion that appeared already after one day of consumption. The surge in thyroid hormone secretion was followed by an adequate decrease in thyroid-stimulating hormone level [65]. It is important to note that the observed phenomenon of thyroid gland activation was not prolonged. Moreover, after two weeks of exposure, the rats developed secondary hypothyroidism due to a sharp decrease in thyroid-stimulating hormone secretion by the pituitary gland. However, by the end of the third week of the experiment, the secretion of hormones increased, and reciprocal dependence between concentrations of thyroid hormones and thyroid-stimulating hormone was restored [66]. Thus, both the thyroid and the hypothalamus–pituitary complex showed some sensitivity to deuterium depletion, but the thyroid gland demonstrated a more rapid response. Investigation of the mechanisms underlying the accelerated secretion of the thyroid gland in response to changes in deuterium content in drinking water revealed an unexpected fact. Increased hormone secretion was not due to activation of resorption and cleavage of thyroglobulin, as usually happens under activating stimulus of the pituitary gland. The main mechanism of the elevated thyroid hormone production was the increase in the synthesis of the Na^+^/I^−^ symporter, a protein forming a channel in the plasma membrane for transportation of the iodides into cells. The Na^+^/I^−^ symporter, as well as other key proteins providing hormone production in the thyroid, is known to be up-regulated by pituitary hormones. In our study, the production of the Na^+^/I^−^ symporter did not demonstrate dependency on the secretion of its main regulator. That is, the change in the deuterium/protium gradient was a regulating factor of Na^+^/I^−^ symporter synthesis that was more powerful than pituitary regulation [66].

Single reports show that deuterium-depleted water consumption can have a positive effect on metabolic disorders, including those caused by endocrine diseases [67,68]. Studies on animals with streptozocin-induced diabetes have shown that consumption of deuterium-depleted water with various D_2_O concentrations does not improve glucose metabolism. But the combination of insulin with 125 ppm deuterium-depleted water produced a more pronounced decrease in glucose and glycated hemoglobin concentrations compared to insulin treatment alone. The researchers revealed that supplementation with deuterium-depleted water significantly enhances insulin’s effect on GLUT-4 (an insulin-dependent glucose transporter) translocation and potentiates glucose uptake in diabetic animals, providing good evidence for its potential use in diabetes care [67]. The regulation of metabolism by lowering the body deuterium content was confirmed by another investigation aimed at evaluating the possible therapeutic effect of deuterium-depleted water on obese rats. The experimental animals had an increased body mass index, glucose concentration, and levels of some pro-inflammatory cytokines; decreased serum insulin levels; and reduced brain tryptophan and serotonin levels. Consumption of deuterium-depleted water at a concentration of 10 ppm for 3 weeks resulted in a decrease in body mass index, serum glucose, increased brain tryptophan and serotonin levels, and an increase in liver zinc concentrations to control levels. A significant increase in the levels of anti-inflammatory interleukin-4 and interleukin-10 was also observed. These data indicate a lower level of systemic inflammation as well as a decrease in oxidative stress and an increase in antioxidant enzyme activity in obese animals treated with deuterium-depleted water [69]. The obtained results can be explained by the effect of reduced body deuterium content on the terminal complex of the mitochondrial electron transfer chain and changes in the rate of fatty acid oxidation and gluconeogenesis, including due to changes in the level of signaling molecules (reactive oxygen species, NO-, and others) [70]. Interestingly, blood glucose concentrations in the obese rats returned to normal levels without changes in insulin secretion, whereas in healthy rats, deuterium-depleted water lowered insulin levels.

Thus, deuterium-depleted water can have a direct effect on endocrine gland secretion and alleviate metabolic disorders caused by endocrine diseases. The analyzed reports show that the response of the endocrine glands controlled by the hypothalamus–pituitary complex and regulated by the concentration of the metabolites to deuterium reduction differs. The above-mentioned results give good reasons for further research of water with modified isotopic compositions and its effects on both the normal functioning of endocrine glands and their functional disorders. The data obtained may form the basis for therapeutic and preventive effects on the endocrine system.

## 7. Possible Mechanisms of Deuterium-Depleted Water Action

Functional changes caused by deuterium-depleted water clearly demonstrate differences between early and late effects and effects on a healthy and diseased organism. Reactive changes that develop during 1–3 days of deuterium-depleted water consumption in a healthy organism, as a rule, are more pronounced. Longer water consumption can change some biochemical and physiological processes as well as return indicators to their initial values. For example, in our studies we found significant reactive changes in the secretion of thyroid hormones and their normalization after one month. The activation of T cell output in the thymus was observed both in the early and late terms, but in the early terms it was much more pronounced. Changes in erythropoiesis were manifested to a greater extent in later terms and involved an increase in hemoglobin synthesis [48,54,58,71]. These changes suggest that a prolonged decrease in deuterium content stimulates cell differentiation to a greater extent than it affects cell metabolism. In the presence of a pathological process in the organism, the changes caused by deuterium-depleted water occur differently. They develop more slowly and last longer [70,72]. Studies conducted on intact animals show that the consumption of deuterium-depleted water by a healthy organism leads to an increase in its adaptive abilities, such as higher hemoglobin content, more active renewal of the T lymphocyte pool, resistance to stress, improved long-term memory, resistance to hypoxia, etc. Consumption of deuterium-depleted water during the development of pathological processes also has a sanogenetic effect. The appearance of a D/H gradient between blood and tissues at the early stages of deuterium level reduction evokes the activation of the nonspecific defense system. This period of preadaptation lasts on average up to 14 days [72,73]. This fact explains the stimulating effect of deuterium reduction in the organism on the immune system, especially in conditions of induced inflammation, shown in various studies. For example, a significantly enhanced inflammatory response to lipopolysaccharide with higher neutrophil and lymphocyte blood content and phagocytic ability of peripheral blood neutrophils was observed in mice receiving deuterium-depleted water with a concentration of 30 ppm [74].

The formation of the isotope gradient between blood and tissues and the enhancement of deuterium exchange for protium play an important role in the initial effects of deuterium-depleted water [27]. The shift in the deuterium/protium ratio in blood, intercellular fluid, and in the cells causes numerous metabolic changes leading to altered physiological parameters. First, there is a change in proton chemiosmosis in the mitochondria and, consequently, an increase in ATP generation. On the other hand, there is also a change in the rates of water and ion transport into the cell and the optimization of redox processes. According to the available data, the altered balance of deuterium and protium affects the number of open states in the DNA molecule, i.e., the sites suitable for transcription [74,75]. In general, this is the most likely mechanism for increasing the production of functional proteins, such as ion channel-forming proteins, and the synthesis of prosthetic groups of enzymes. In turn, the increase in the formation of ion channels and enzyme activity allows us to improve the absorption of nutrients such as iron, iodine, glucose, etc., and these changes can influence morphogenetic processes such as cell proliferation and differentiation. It is possible that some of these processes are also related to the occurrence of the magnetic isotope effect, since long-term consumption of deuterium-depleted water leads to an increase in the content of the magnetic isotope protium in cells. In Figure 2, we present the most likely mechanism of the physiological effects of deuterium-depleted water leading to an increase in the organism’s adaptive capacity.

Considering the pros and cons of deuterium-depleted water as a therapeutic tool, it is very important to assess the potential risk of prolonged deuterium depletion. Of particular concern is the activation of cell proliferation. This phenomenon can both compensate for the decline and deficiency of some cell types and promote the development of hyperplasias, especially if the organism already has a latent similar process. The impact of reduced deuterium content on cell migration is poorly studied and is also of concern because retention of actively proliferating cells like hemopoietic progenitors increases the risk of malignant transformation.

## 8. Conclusions

The analyzed data convince us that changes in the hydrogen isotopic composition of the main nutrient water do initiate changes in organ functioning and provide therapeutic and preventive effects in various pathological conditions. Consequently, the isotopic composition of water should be considered a regulator of physiological processes, able to be used as a powerful preventive and therapeutic tool. The ability to change the isotopic composition of consumed water to enhance the assimilation of other nutrients and regulate the balance of cell proliferation and differentiation allows us to revise traditional treatment schemes, including diseases associated with the deficiency of certain nutrients, with the decrease in metabolic activity and defense reactions with age, and with the increase in adaptive capabilities of the organism. The use of deuterium-depleted water as a corrector of disorders caused by anthropogenic stress and occupational hazards is extremely understudied. At the same time, it should be recognized that short-term and long-term water consumption may have different effects, and therefore deuterium-depleted water may be administered in short courses to induce metabolic changes. The unexplored nature of many effects of deuterium depletion in the organism necessitates cautious use of deuterium-depleted water and avoidance of its uncontrolled intake. New data on the mechanisms of action of deuterium-depleted water open directions for future research, both theoretical and applied. However, in these studies it is necessary to pay much attention not only to changes in functional parameters and the duration of these changes, but also to changes in morphogenetic processes in various tissues and organs and physiological self-maintenance of cell populations, which is the least studied aspect. Studying these issues will enable future development of preventive and therapeutic concepts using deuterium-depleted water.

## Figures and Tables

**Figure 1 ijms-26-04448-f001:**
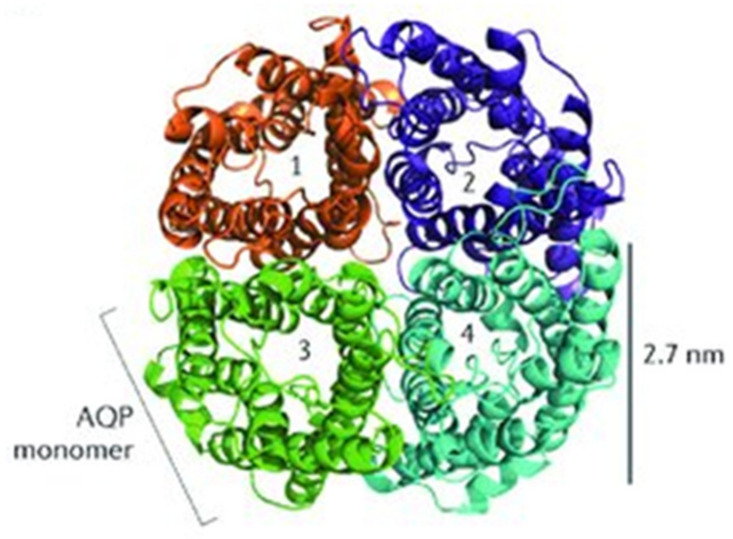
The quaternary organization of the AQP1 [14].

**Figure 2 ijms-26-04448-f002:**
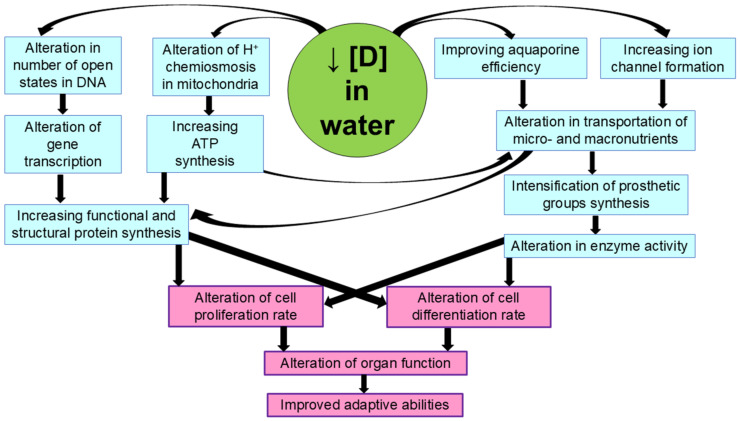
Possible mechanisms of the physiological effects of deuterium-depleted water.

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
