# Peer review of "Altering the Hydrogen Isotopic Composition of the Essential Nutrient Water as a Promising Tool for Therapy: Perspectives and Risks"

_ijms, 2025, doi:10.3390/ijms26094448_

Round 1
Reviewer 1 Report
Comments and Suggestions for Authors
Dear authors,
We would like to thank you for a good review article. Keep up the good work.
Page 1
The title and abstract are okay.
The introduction is too short. One paragraph intro is okay with you? Please add a few more paragraphs with proper citations.
Page 2
You wrote (Several isotopic forms of water are present in the living organisms [1,2]. The most common is 1H216O.), the 2 under the H is not superscript.
It should read:
(Several isotopic forms of water are present in the living organisms [1,2]. The most common is 1H216O.)
Lines 56 and 57, you wrote (Semi-heavy water (HDO or 1H22H216O) is the most common form of deuterated water in nature [2,5,6].)
There are two issues here:
1-
It should read (HDO or 1H2H16O) no need to have the 2 as a subscript. Look at the difference!
The super/superscripted font is making you dizzy. You need to go over the entire manuscript and make sure that all the notations are correct
2- Do you really need a new paragraph that contains 1.3 lines?
Merge this line (fact) with the previous paragraph or the next paragraph.
In Line 59, you wrote 18O instead it should read 18O instead. This is the last subscript/superscript text I am going to correct.
PLEASE GO OVER THE ENTIRE MANUSCRIPT AND FIX any text that needs fixing.
Line 77, you need to fix the number (to be extremely high, ∼3×109 water molecules) to read (to be extremely high, ∼3×109 water molecules)
Between lines 102-111, you did not give the proper references that back up your claims in this lengthy paragraph. Few references are needed with this paragraph to back up your writing.
We liked the fact that you wrote in line 217, (deuterium-depleted water promoted the growth of normal cells.) However, it is better if you cite the proper reference that cit this fact.
Overall, the manuscript has authentic writings with proper references.
Figures: Only one schematic figure presented. It looked good.
Tables: No tables were presented.
Conclusion:
The conclusion presented is appropriate, however, it looks and sounds similar to the abstract presented at the top of the manuscript. Please rewrite.
Also, it sounds like an introduction.
For example, you wrote (Water is the most important nutrient, and its quality and properties are determined not only by mineral and microbiological composition, but also by isotopic composition. Changes in the hydrogen isotopic composition do initiate changes in organs functioning and provide therapeutic and preventive effects in various pathological conditions.)
This is very similar to an introduction to a paper.
References:
There are 7 articles by the corresponding author. This is considered improper self-citation per the guidelines of IJMS. Please, make it one (1) or two self-cited references.
Please go over every single reference and check that the format is consistent and you have inserted the proper doi number. Such as reference 33 has no doi number at the end of the reference.
Author Response
Please see the attachement.

Reviewer 2 Report
Comments and Suggestions for Authors
In the manuscript, the authors mainly focused on introducing the function of water as a promising tool for different diseases therapy. The authors not only introduced the background, the mechanisms, but also the present associated studies. In fact, the field of altering hydrogen isotopic composition of the essential nutrient water is not known by researchers present. The review manuscript expands the possibility of disease therapy using water. Besides, the workload of the manuscript is enough. Therefore, considering the meaning and the workload of the manuscript, the work is recommended to be accepted by the International Journal of Molecular Sciences. However, as far as I am concerned, I think the authors should improve the quality of the manuscript as follows:
1) The authors should revise the start of the last sentence in the Abstract section as research subject should be the subject.
2) In the Keywords section, the 2nd and the 3rd word should be deleted, and “water” should be added as the first keyword.
3) In the 2nd paragraph of section 3, the authors mentioned that AQP1 was found to exist as a tetramer with intracellular N and C termini. The authors should add the 3D picture of the protein to make it understand better.
4) In the whole manuscript, the authors lost a blank space between the number and its unit. The authors are recommended to revise such error.
5) Sometimes, the authors type the subject “deuterium-depleted water”, but sometimes “deuterium depleted water”. The authors should recheck such inconsistencies and use the same format.
6) In the 2nd paragraph of section 6.3, the authors mentioned their research work. Can the authors add their associated reference?
7) I recommend the authors add one graphic figure like one workflow to summarize the whole contents of the review work to help readers understand (less word but more pictures is recommended).
Round 2
Reviewer 2 Report
Comments and Suggestions for Authors
The authors have revised the manuscript better. And it was recommended to be accepted by the International Journal of Molecular Sciences. Congratulationals to the authors.